# Kampo (Traditional Japanese Herbal) Formulae for Treatment of Stomatitis and Oral Mucositis

**DOI:** 10.3390/medicines5040130

**Published:** 2018-12-10

**Authors:** Masataka Sunagawa, Kojiro Yamaguchi, Mana Tsukada, Nachi Ebihara, Hideshi Ikemoto, Tadashi Hisamitsu

**Affiliations:** Department of physiology, School of medicine, Showa University, Tokyo 142-8555, Japan; suna@med.showa-u.ac.jp (M.S.); kampo5260@icloud.com (K.Y.); m-tsukada@med.showa-u.ac.jp (M.T.); BYS00426@nifty.com (N.E.); h_ikemoto@med.showa-u.ac.jp (H.I.)

**Keywords:** kampo formula, traditional Japanese herbal medicine, stomatitis, mucositis, Hangeshashinto

## Abstract

Stomatitis is occasionally multiple, recurrent, and refractory. Currently, mucositis induced by chemotherapy and radiation therapy in patients with cancer has become a significant clinical problem. Effective treatments have not been established and the treatment of numerous cases remains a challenge for physicians. Traditional Japanese herbal medicines termed Kampo formulae (i.e., Hangeshashinto, Orengedokuto, Inchinkoto, Orento, Byakkokaninjinto, Juzentaihoto, Hochuekkito, and Shosaikoto) are used for treating various types of stomatitis and mucositis. Its use has been based on the Kampo medical theories—empirical rules established over thousands of years. However, recently, clinical and basic research studies investigating these formulae have been conducted to obtain scientific evidence. Clinical studies investigating efficacies of Shosaikoto and Orento for the treatment of cryptogenic stomatitis and acute aphthous stomatitis and those investigating the effects of Hangeshashinto, Orengedokuto, and Juzentaihoto on chemotherapy- or radiotherapy-induced mucositis have been conducted. The Kampo formulae comprise several crude drugs, whose mechanisms of action are gradually being clarified. Most of these drugs that are used for the treatment of stomatitis possess anti-inflammatory, analgesic, and antioxidative properties. In this review, we introduce the clinical applications and summarize the available evidence on the Kampo formulae for the treatment of stomatitis and oral mucositis.

## 1. Introduction

Stomatitis is a sore and often recurrent inflammatory condition of the oral mucosa, characterized by various symptoms such as the presence of vesicles, erosions, aphthae, and ulcerations. Stomatitis is caused by various factors such as viral, fungal, and bacterial infections, allergic reactions, loose-fitting dental prosthetics, and systemic diseases. Occasionally, stomatitis is multiple, recurrent, and refractory. Currently, mucositis induced by chemotherapy and radiation therapy in patients with cancer has become a significant clinical problem [1]. The pain associated with mucositis often affects a patient’s functional status and quality of life.

Kampo formula, a traditional Japanese herbal medicine, has its root in ancient Chinese medicine, and the antecedent form of medicine was introduced to Japan between the 5th and 6th century. It was developed into an individual form of medicine adapting the constitutions of the Japanese people. Kampo formulae have been reported to be effective for the treatment of stomatitis and mucositis [2,3]. The objective of this review was to introduce the clinical applications and summarize the available evidence on the Kampo formulae for the treatment of these two conditions.

## 2. Clinical Applications

We conducted a questionnaire survey regarding the treatment of stomatitis using the Kampo formulae. According to the results, formulae such as Hangeshashinto (HST), Orengedokuto (OGT), Inchinkoto (ICT), Orento (ORT), Byakkokaninjinto (BKN), Juzentaihoto (JTT), Hochuekkito (HET), and Shosaikoto (SST) were used (Figure 1) [4]. The Kampo formulae are generally composed of at least two kinds of crude drugs and these combinations may suppress the development of infections, inflammation, concomitant oxidative stress, and the underlying causes of stomatitis. The chief ingredient and principal effects of each crude drug included in the formulae frequently used for the treatment of stomatitis are shown in Table 1. Many of these agents possess anti-inflammatory and/or analgesic properties. Of note, *Astragali* Radix, *Scutellariae* Radix, *Phellodendri* Cortex, *Coptidis* Rhizoma, *Glycyrrhizae* Radix, *Bupleuri* Radix, *Paeoniae* Radix, *Artemisiae Lanceae* Rhizoma, *Cimicifugae* Rhizoma, *Cnidii* Rhizoma, *Angelicae* Radix, and *Poria* exert anti-inflammatory effects. Moreover, *Cinnamomi* Cortex, *Cimicifugae* Rhizoma, *Paeoniae* Radix, *Glycyrrhizae* Radix, *Cnidii* Rhizoma, *Angelicae* Radix, *Zingiberis* Rhizoma Processum, and *Magnoliae* Cortex exert analgesic effects [5]. The occurrence of stomatitis is related to the generation of reactive oxygen species (ROS) [6]. Therefore, anti-oxidants contained in these medicinal herbs may effectively mitigate this damaging effect. Dragland et al. [7] assessed the contribution of culinary and medicinal herbs to the total dietary intake of anti-oxidants (Table 1). Notably, *Cinnamomi* Cortex, *Scutellariae* Radix, *Cimicifugae* Rhizoma, *Paeoniae* Radix, *Aurantii Nobilis* Pericarpium, and *Glycyrrhizae* Radix contain high concentrations of anti-oxidants.

In addition, *Cinnamomi* Cortex, *Scutellariae* Radix, *Glycyrrhizae* Radix, *Astragali* Radix, *Coptidis* Rhizoma, and *Phellodendri* Cortex inhibit several bacterial infections. Moreover, *Cinnamomi* Cortex, *Scutellariae* Radix, and *Anemarrhenae* Rhizome exert antifungal effects. Furthermore, *Scutellariae* Radix and *Glycyrrhizae* Radix have been shown to inhibit viral infections [8].

In Kampo medicine, the approach to the treatment of stomatitis differs depending on the nature of the symptoms (i.e., acute or chronic). Furthermore, the most appropriate treatment is determined according to the presence of oral and general symptoms. The effectiveness of each Kampo formula according to the pathognomonic symptoms of patients is shown in Table 2. In the acute type, formulae exerting cooling effects (i.e., HST, ORT, OGT, BNT, SST, and ICT) are used. In the chronic type, formulae such as HIS, JTT, and HET are used [3]. The JTT and HET supply energy in patients with symptoms of tiredness, fatigue, or lowered/suppressed immunity [9,10]. The oral cavity is a part of the digestive system; thus, digestive symptoms are important selection criteria for the Kampo formulae. The HST, ORT, OGT, SST, ICT, HIS, JTT, and HET are applied for the treatment of gastrointestinal diseases and symptoms by the national health insurance in Japan.

## 3. Clinical Studies

Use of the Kampo formulae has been based on the Kampo medical theories—empirical rules established over thousands of years. In recent years, clinical and basic research studies investigating the Kampo formulae have been performed to obtain scientific evidence.

For our literature review, PubMed (National Center for Biotechnology Information, Bethesda, MD, USA) and ICHUSHI (Japan Medical Abstracts Society, Tokyo, Japan) were used to identify relevant evidence. Reports of clinical studies (randomized controlled trials, case-control studies, and case series studies) identified through this search are summarized in Table 3.

Ogino, et al. [11] showed that the efficacy of SSK administered to patients with cryptogenic stomatitis accompanied by pain (n = 10) was 80%. SSK was particularly effective against symptoms such as erosion and redness. Oka [12] investigated the effect of ORT in patients with acute aphthous stomatitis (n = 39). The numbers of days until the resolution of pain (2.6 days) and complete cure (6.3 days) were reduced in patients treated with ORT compared with those observed in patients treated with a steroid ointment (pain: 7.5 days; cure: 12.3 days).

Currently, there are effective treatment options for chemotherapy- or radiotherapy-induced stomatitis. HST exerts a preventive effect against these types of stomatitis. Yuki, et al. [13] administered OGT for the treatment of chemotherapy-induced stomatitis and diarrhea in patients with acute myeloblastic or lymphoblastic leukemia (n = 40). The incidence of stomatitis in ORG-treated patients (n = 14) was significantly lower compared with that reported in those who received a gargle consisting of allopurinol, sodium gualenate, and povidone-iodine (n = 25) (27.9% vs. 71.6%, respectively; *p* < 0.0001). Moreover, the incidence of diarrhea was significantly lower (9.3% vs. 31.7%, respectively; *p* < 0.005). HST has been administered for the treatment of chemotherapy-induced mucositis in patients with various types of cancers, such as colorectal [14,15,16,17], gastric [16,17,18], and renal cancers [19]. In all studies, HST extract granules (TJ-14; Tsumura, Tokyo, Japan) were dissolved in drinking water and subjects rinsed their oral cavity with the solution thrice daily. Kono, et al. [14] reported that 92.8% of patients (13/14 patients) with chemotherapy-induced oral mucositis (COM) during treatment with mFOLFOX6 or FOLFIRI for metastasis of advanced colorectal cancer showed significant improvement following a 1-week topical application of HST. Moreover, a significantly decreased mean Common Terminology Criteria for Adverse Events grade was reported in patients treated with HST (*p* = 0.0012). Aoyama, et al. [18] conducted a double-blinded, placebo-controlled, randomized study of HST for the treatment of COM in patients with gastric cancer (n = 91). Although treatment with HST did not reduce the incidence of grade ≥ 2 COM, a trend toward the reduction of the risk of grade 1 COM by HST was observed during the screening cycle. Matsuda, et al. [15] also conducted a double-blinded, randomized study investigating the effect of HST against mucositis induced by infusional fluorinated-pyrimidine-based colorectal cancer chemotherapy (n = 93). Although the incidence of grade ≥ 2 mucositis was lower in patients treated with HST than in those treated with placebo, the difference was not statistically significant (48.8% vs. 57.4%, respectively; *p* = 0.41). The median duration of grade ≥ 2 mucositis was 5.5 days versus 10.5 days, respectively (*p* = 0.018). Nishikawa, et al. [17] demonstrated similar results in patients with gastric and colorectal cancer (n = 181), with a median time to improvement from grade ≥ 2 to grade < 1 COM of 8 days versus 15 days in the HST and placebo groups, respectively (*p* = 0.072). Yoshida, et al. [16] and Ohoka, et al. [19] also administered HST to patients with various types of cancer, demonstrating significant decreases in the Common Terminology Criteria for Adverse Events (v4.0) grades. The findings of these studies suggested that HST may be effective for the treatment of chemotherapy- or radiotherapy-induced stomatitis.

A previous study investigated the administration of JTT for the treatment of radiation-induced stomatitis in patients with oral cancer, in whom oral ingestion was not possible (n = 15) [20]. The mean period during which oral ingestion was not possible in these patients showed a reducing trend (i.e., 17.9 ± 7.1 days vs. 26.0 ± 11.6 days in the JTT-treated (n = 8) and non-treated (n = 7) groups, respectively (*p* < 0.121)). The clinical use of HST for the treatment of radiation-induced stomatitis has also been reported [21,22].

## 4. Basic Studies of HST

Stomatitis and oral mucositis are induced by various factors, such as infection, inflammation, concomitant oxidative stress, suppressed immunity, depressed function of the digestive tract, malnutrition, psychological stress, and physical stress. Regarding chemotherapy- and radiotherapy-induced mucositis, these treatments induce DNA and non-DNA damage that results in injury of basal epithelial, submucosal, and endothelial cells. In response to this damage, oxidative stress results in the formation of ROS. The presence of ROS damages cell membranes, induces proinflammatory cytokines such as tumor necrosis factor-α, prostaglandin (PG) E_2_, interleukin-6, and interleukin-1β, and upregulates cyclooxygenase (COX)-2 in submucosal fibroblasts and endothelial cells leading to mucosal ulceration [23,24]. Furthermore, chemotherapy and radiotherapy reduce immunity, facilitating the development of infectious diseases [25].

Hitomi, et al. [26] conducted in-vivo studies evaluating the analgesic effects of HST using an oral ulcer rat model treated with acetic acid. The topical application of HST in ulcerative oral mucosa suppressed mechanical pain hypersensitivity without exerting effects on healthy mucosa. Moreover, Kamide, et al. [27] assessed the effectiveness of HST for the prevention of radiation-induced mucositis using a hamster model. Administration of HST significantly reduced the severity of mucositis. The percentage of severe mucositis (score ≥ 3) was 100% and 16.7% in the untreated and HST groups, respectively. Moreover, HST inhibited the infiltration of neutrophils and expression of COX-2 in irradiated mucosa.

As mentioned earlier in this review, the main characteristic of the Kampo formulae is the combination of several crude drugs (Table 1). These combinations may suppress multiple causes of stomatitis and mucositis. Of note, HST is composed of seven herbs, namely *Scutellariae* Radix, *Glycyrrhizae* Radix, *Zizyphi* Fructus, *Ginseng* Radix, *Pinelliae* Tuber, *Zingiberis* Rhizoma Processum, and *Coptidis* Rhizoma. These constituents, except *Pinelliae* Tuber, exert antioxidative effects [7,28]. Matsumoto, et al. [28] reported that *Glycyrrhizae* Radix, *Ginseng* Radix, and *Zizyphi* Fructus demonstrated scavenging activity for hydroxyl radical, while *Scutellariae* Radix, *Glycyrrhizae* Radix, *Zingiberis* Rhizoma Processum, and *Coptidis* Rhizoma eliminated superoxide. Moreover, *Scutellariae* Radix and *Coptidis* Rhizoma have been shown to eliminate nitroxyl radical.

Regarding its antibacterial action, in vitro studies demonstrated that HST extract inhibited the growth of Gram-negative bacteria including Fusobacterium nucleatum, Porphyromonas gingivalis, Porphyromonas endodontalis, Prevotella intermedia, Prevotella melaninogenica, Tannerella forsythia, Treponema denticola, and Porphyromonas asaccharolytica. However, these inhibitory effects were less pronounced in Gram-positive bacteria and Candida albicans. These effects are thought to be induced by *Scutellariae* Radix (baicalein), *Pinelliae* Tuber (homogentisic acid), *Zingiberis* Rhizoma Processum ([6]-shogaol) and *Coptidis* Rhizoma (berberine, coptisine) [29]. Furthermore, *Glycyrrhizae* Radix [30,31], *Pinelliae* Tuber [31], *Coptidis* Rhizoma [32], and *Ginseng* Radix may enhance immunity. In particular, *Ginseng* Radix was reported to increase the activity of natural killer cells in mice treated orally with it [33,34].

Regarding its anti-inflammatory effect, HST inhibited the production of PGE_2_ and suppressed the expression of COX-2 protein. In vitro studies show that *Scutellariae* Radix, *Glycyrrhizae* Radix, *Zingiberis* Rhizoma Processum, and *Coptidis* Rhizoma are involved in these effects [35,36,37,38]. As stated earlier in this review, Hitomi, et al. reported the analgesic action of HST [26] and subsequently found through in vitro and in vivo studies that blockage of Na^+^ channels by components of *Zingiberis* Rhizoma Processum ([6]-gingerol and ([6]-shogaol) play an essential role in HST-associated analgesia. Moreover, the *Ginseng* Radix extract demonstrated an acceleration of substance permeability into the tissue of the oral ulcer and enhanced the analgesic action of *Zingiberis* Rhizoma Processum [39]. Baicalein—an active constituent of *Scutellariae* Radix—alleviated mechanical allodynia in rats with cancer-induced bone pain [40]. Glycyrrhizin—an active constituent of *Glycyrrhizae* Radix—ameliorated inflammatory pain by inhibiting the microglial activation-mediated inflammatory response in mice with inflammatory pain [41]. HST induces its analgesic effect through the synergistic actions of certain crude drugs. Therefore, HST exerts a combination of antioxidative, anti-inflammatory, immunostimulatory, and analgesic effects. Moreover, HST is able to control the symptoms and simultaneously eliminate the underlying causes of the condition (Figure 2).

## 5. Conclusions

In general, western medicines such as steroid ointments, nonsteroidal anti-inflammatory drugs, and antiviral and antifungal drugs are applied for stomatitis and oral mucositis according to the causes and symptoms. In contrast to western medicines which generally include a single component, the Kampo formulae contain multiple components and their effects are exerted through complex mechanisms of action. Use of the Kampo formulae may be an alternative treatment option for patients who failed to respond to conventional therapies. In addition, the concomitant use of Kampo formulae with western medicines may be useful. Kobayashi [42] reported effective cases in which the concomitant use of HST with steroid ointment was applied. In this review, we introduced the clinical applications and summarized the available evidence of the Kampo formulae for the treatment of stomatitis and oral mucositis. Despite the availability of clinical reports, the evidence (except for that related to treatment with HST) is limited. Future clinical and basic research studies are warranted to further investigate the effectiveness on the Kampo formulae against these conditions.

## Figures and Tables

**Figure 1 medicines-05-00130-f001:**
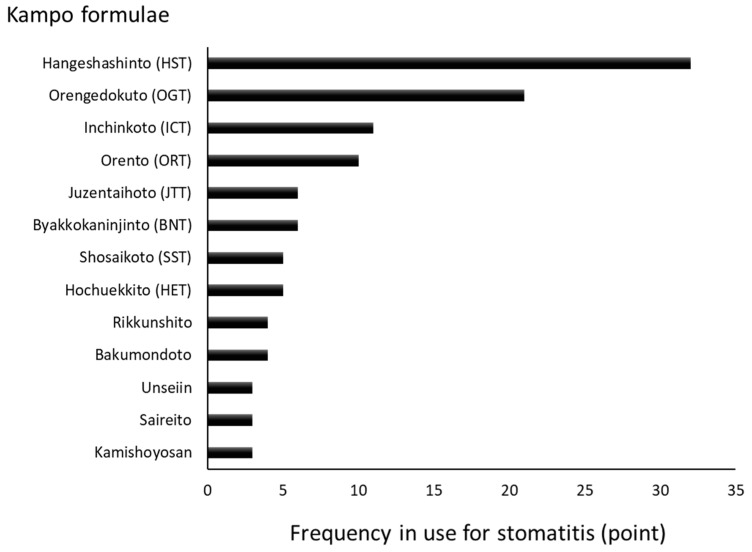
Kampo formulae frequently used for the treatment of stomatitis [4]. The trends in the use of the Kampo formulae at hospitals and faculties of oral surgery of dental/medical universities in Japan were surveyed. A total of 55 hospitals participated in the survey and rated the frequency of Kampo formulae use via a scale from 0 to 3. This graphic summarizes the results of rating.

**Figure 2 medicines-05-00130-f002:**
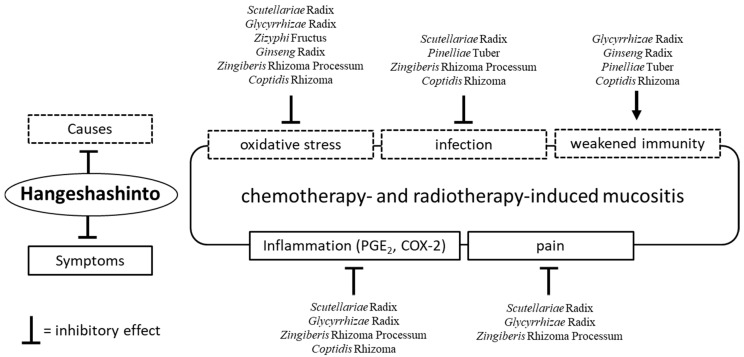
The effects of Hangeshashinto (HST) and the crude drugs on chemotherapy- and radiotherapy-induced mucositis. HST exerts antioxidative, anti-inflammatory, immunostimulatory, and analgesic effects. Moreover, HST is able to control the symptoms and simultaneously eliminate the underlying causes of the condition. PGE_2_; prostaglandin E_2_. COX-2; cyclooxygenase-2.

**Table 1 medicines-05-00130-t001:** The Kampo formulae frequently used for the treatment of stomatitis and their corresponding crude drugs.

		Kampo Formulae											
Crude Drug/Japanese Name		Hangeshashinto (HST)	Orento (ORT)	Orengedokuto (OGT)	Byakkokaninjinto (BKN)	Shosaikoto (SST)	Inchinkoto (ICT)	Heiisan (HIS)	Juzentaihoto (JTT)	Hochuekkito (HET)	Antioxidant [7] (mmol/100 g)	Chief Ingredient	Principal Effects
*Cinnamomi* Cortex	Keihi		◯			◯			◯		120.2	cinnamaldehyde	antipyresis, perspiration, analgesia
*Scutellariae* Radix	Ogon	◯		◯		◯					111.5	baicalin	anti-inflammation, antipyresis, laxative
*Cimicifugae* Rhizoma	Shoma									◯	64.3	cimigenol	anti-inflammation, antipyresis, analgesia, antiedema
*Paeoniae* Radix	Shakuyaku					◯			◯		55.1	paeoniflorin	analgesia, spasmolysis, anti-inflammation
*Aurantii Nobilis* Pericarpium	Chinpi							◯		◯	17.5	hesperidin	stomachic, antitussive
*Glycyrrhizae* Radix	Kanzo	◯	◯		◯	◯		◯	◯	◯	11.6	glycyrrhizin	anti-inflammation, analgesia, detoxification
*Zingiberis* Rhizoma	Shokyo					◯		◯		◯	7.5	gingerol	stomachic, antinausea
*Atractylodis Lanceae* Rhizoma	Sojutsu							◯	◯	◯	7.4	atractylodin	anti-inflammation, stomachic, diuresis
*Cnidii* Rhizoma	Senkyu								◯		6.7	cnidilide	analeptic, nourishing, anti-inflammation, analgesia
*Zizyphi* Fructus	Taiso	◯	◯			◯		◯		◯	5.9	zizyphus saponin	analeptic, nourishing, stomachic
*Bupleuri* Radix	Saiko					◯				◯	5.7	saikosaponin	anti-inflammation, antipyresis
*Astragali* Radix	Ogi								◯	◯	4.9	formononetin	anti-inflammation, analeptic, diuresis, hypotensive
*Rhemanniae* Radix	Jio								◯		3.9	catalpol	nourishing, diuresis
*Angelicae* Radix	Toki								◯	◯	3.0	ligustilide	analeptic, nourishing, anti-inflammation, analgesia
*Hoelen*	Bukuryo								◯		2.8	eburicoic acid	antiedema, stomachic
*Ginseng* Radix	Ninjin	◯	◯		◯	◯			◯	◯	1.5	ginsenoside	stomachic, nourishing, antinausea
*Pinelliae* Tuber	Hange	◯	◯			◯					0.3	homogentisic acid	sedation, antinausea, antitussive
*Zingiberis* Rhizoma Processum	Kankyo	◯	◯									shogaol	warming, analgesia
*Coptidis* Rhizoma	Oren	◯	◯	◯								berberine	anti-inflammation, stomachic, antibacterial, spasmolysis
*Phellodendri* Cortex	Obaku			◯								berberine	anti-inflammation, stomachic
*Gardeniae* Fructus	Sanshishi			◯			◯					geniposide	anti-inflammation, antipyresis, choleresis
*Artemisiae* Capillaris Flos	Inchinko						◯					capillarisin	anti-inflammation, antipyresis, choleresis
*Rhei* Rhizoma	Daio						◯					sennoside	laxative, blood fluidity improving
*Magnoliae* Cortex	Koboku							◯				magnolol	stomachic, analgesia, spasmolysis
*Gypsum*	Sekko				◯							calcium sulfate	anti-inflammation, antipyresis, sedation
*Oryzae* Fructus	Kobei				◯							starch	stomachic, nourishing
*Anemarrhenae* Rhizoma	Chimo				◯							timosaponin AIII	antipyresis, hypoglycemia

Regarding the antioxidative effects of these formulae, only herbs that Dragland, et al. [7] assessed are described.

**Table 2 medicines-05-00130-t002:** The effectiveness of each Kampo formula according to the pathognomonic symptoms of patients.

Kampo Formulae	Pathognomonic Symptoms
Hangeshashinto (HST)	multiple stomatitis irritation, anxiety, insomnia, rush of blood to the head, anorexia, diarrhea, epigastric discomfort and resistance
Orento (ORT)	multiple stomatitis rush of blood to the head, anorexia, decrease in digestive function, abdominal chill symptom, abdominalgia due to chill, epigastric discomfort and resistance
Orengedokuto (OGT)	multiple stomatitis irritation, insomnia, rush of blood to the head
Byakkokaninjinto (BKN)	thirstiness, dry mouth hyperidrosis, polyuria
Shosaikoto (SST)	bitter in the mouth irritation, depression, anorexia, hypochondriac discomfort and distension, nausea
Inchinkoto (ICT)	multiple stomatitis, dry mouth irritation, insomnia, constipation, oliguria
Heiisan (HIS)	multiple stomatitis anorexia, decrease in digestive function, abdominal distension
Juzentaihoto (JTT)	chronic and repetitive stomatitis, dry mouth depressed, fatigue, dullness, macies, hot sensation, night sweat, anemia, anorexia, decrease in digestive function
Hochuekkito (HET)	chronic and repetitive stomatitis depressed, fatigue, dullness, anorexia, decrease in digestive function

**Table 3 medicines-05-00130-t003:** Clinical studies reporting the use of the Kampo formulae for the treatment of stomatitis.

No.	First Author, Year [Reference No.]	Kampo Formula	Study Design	Target Patient	Principal Result
1	Ogino, 1992 [11]	Shosaikoto (SSK)	case series study	cryptogenic stomatitis (n = 10)	Efficacy rate was 80%. (very effective = 2, effective = 4, slightly effective = 2, no change = 2)
2	Oka, 2007 [12]	Orento (ORT)	RCT	acute aphthous stomatitis (n = 39) > non-treated (n = 6), steroid ointment-treated (n = 6) and ORT-treated (n = 27) groups	The administration of Orento reduced the number of days until the disappearance of pain and the complete cure compared to other groups.
3	Yuki, 2003 [13]	Orengedokuto (OGT)	case-control (retrospective) study	chemotherapy-induced stomatitis in patients with acute leukemia (n = 40) > ORG-treated (n = 15) and gargling (n = 25) groups	Incidence of stomatitis was 27.9% in the ORG-treated group, which was significantly lower compared with 71.6% in those who received a gargle consisting of allopurinol, sodium gualenate, and povidone-iodine (*p* < 0.0001).
4	Kono, 2010 [14]	Hangeshashinto (HST)	case series study	chemotherapy-induced oral mucositis during mFOLFOX6 or FOLFIRI treatment for metastasis of advanced colorectal cancer (n = 14)	Thirteen patients (92.8%) showed improvements in oral mucositis, with significantly decreased mean CTCAE grades (*p* = 0.0012).
5	Aoyama, 2014 [18]		RCT	gastric cancer chemotherapy-induced oral mucositis (COM) (n = 91) > HST-treated (n = 45) and placebo (n = 46) groups	Although HST treatment did not reduce the incidence of ≥grade 2 COM, a trend was observed in which HST reduced the risk of COM in the patients who developed grade 1 COM.
6	Matsuda, 2015 [15]		RCT	infusional fluorinated-pyrimidine-based colorectal cancer chemotherapy-induced oral mucositis (n = 93) > HST-treated (n = 46) and placebo (n = 47) groups	Although the incidence of grade ≥2 mucositis was lower for patients treated with HST compared to those treated with placebo, there was no significant difference (48.8 vs. 57.4%; *p* = 0.41). The median duration of grade ≥2 mucositis was 5.5 versus 10.5 days (*p* = 0.018).
7	Yoshida, 2017 [16]		case series study	cancer chemotherapy-induced oral mucositis (grade ≥ 2) (n = 50)	Thirty-seven patients (74%) showed improvements in oral mucositis, with significantly decreased mean NRS and CTC-grade (*p* < 0.001).
8	Nishikawa, 2018 [17]		RCT	chemotherapy-induced oral mucositis (COM) in patients with gastric cancer and colorectal cancer (n = 181) > HST-treated (n = 88) and placebo (n = 93) groups	The incidence of grade ≥2 COM in the HST group was 55.7%, while that in the placebo group was 53.8% (*p* = 0.796). The median time to remission of grade ≥ 2 COM to grade < 1 was 8 days in the HST group and 15 days in the placebo group (*p* = 0.072).
9	Ohoka, 2018 [19]		RCT	sunitinib-induced oral mucositis (OM) in patients with metastatic renal cancer (n = 22) > HST-gargling (n = 12) and non-gargling (n = 10) groups	The gargling with HST significantly improved OM grade and eating status (Global self assessment) (*p* = 0.002).
10	Wada, 2004 [20]	Juzentaihoto (JTT)	RCT	radiation (40 Gy >)-induced stomatitis in patients with oral cancer (n = 15) > JTT-treated (n = 8) and non-treated (n = 7) groups	The mean period that patients could not ingest orally was 17.9 ± 7.1 days in the JTT-treated group, while that in the non-treated group was 26.0 ± 11.6 day (*p* = 0.121).

RCT: randomized controlled trial.

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
