# Peer review of "Kampo (Traditional Japanese Herbal) Formulae for Treatment of Stomatitis and Oral Mucositis"

_medicines, 2018, doi:10.3390/medicines5040130_

Reviewer 1 Report

Since the use of  medicinal herbs has been a great success in the last few decades, this  article is of great interest in dentistry. This paper meet a high standard of scientific quality and credibility.  It is readable and appropriately presented, and  contains appropriate referencing and any recognizable plagiarism. This paper Is compliant with the aims and scope of the journal and  meet ethical requirements.

Author Response

I would like to thank you for your review.

Reviewer 2 Report

Dear Authors,

Many congratulations on the successful compilation and well-structured presentation of the contribution of japanese Kampo formulae to treatment of chemo and radiotherapy induced stomatitis and oral mucositis. The manuscript titled ''Kampo (traditional Japanese herbal) formulae for treatment of stomatitis and oral mucositis", has documented the contibution of Kampo formulae to stomatitis and oral mucositis with appropriate published literature. The manuscript is well structured and the tabulated information adds strength and clarity to the content presented. Although the manuscript is presented well, there are a few minor concerns which need to be addressed in order for the manuscript to be accepted for publication in Medicines. 

1. In the Introduction section, the authors could add a brief description for kampo medicine to introduce the readers to kampo medicine. 

2. In figure 1, the x axis and y axis could have titles. Especially, the X-axis points have to be explained in the figure legend. 

3. The font size of the contents in Table 1, are too small to be readable. If there is away to make it more readable, that has to be done.

4. In figure 2, the two arrows originating from HST need to be changed to inhibitory symbols, because the text in the manuscript states that HST is capable of controlling symptoms and eliminating causes (which also could be denoted with a X mark). Likewise, the arrows pointing towards inflammation, pain, oxidative stress and infection have to be replaced with inhibitory symbols and for inflammation PGE2 could as well be mentioned on the figure. 

5. In Conclusion, the authors state that 'In addition, the concomitant use of Kampo formulae with western medicines may be useful'. It is important that the authors have to clarify whether, concomitant use of kampo medicine with western medicine has been attempted or not in the past.   

Author Response

I would like to thank you for your review. I have revised my manuscript as follows.

1. In the Introduction section, the authors could add a brief description for kampo medicine to introduce the readers to kampo medicine.

→ I added the explanation. (Page 1, L36)

2. In figure 1, the x axis and y axis could have titles. Especially, the X-axis points have to be explained in the figure legend.

→ The titles and the explanation of ‘point’ were inserted.

3. The font size of the contents in Table 1, are too small to be readable. If there is a way to make it more readable, that has to be done.

→ The font size was made larger (also Table 2 & 3). 

4. In figure 2, the two arrows originating from HST need to be changed to inhibitory symbols, because the text in the manuscript states that HST is capable of controlling symptoms and eliminating causes (which also could be denoted with a X mark). Likewise, the arrows pointing towards inflammation, pain, oxidative stress and infection have to be replaced with inhibitory symbols and for inflammation PGE2 could as well be mentioned on the figure.

→ Figure 2 was modified.

5. In Conclusion, the authors state that 'In addition, the concomitant use of Kampo formulae with western medicines may be useful'. It is important that the authors have to clarify whether, concomitant use of kampo medicine with western medicine has been attempted or not in the past.

→ The example was inserted in conclusion.

Reviewer 3 Report

Good review. In the section on Basic studies of HST it should be clear on various statements the effects of HST if it is an animal study or in-vitro or human trial used to determine the mentioned action.Also when discussing western medicines you  should state which medicines . This will make it a better paper for the readers.

Author Response

I would like to thank you for your review. I have revised my manuscript as follows.

In the section on Basic studies of HST, it should be clear on various statements the effects of HST if it is an animal study or in-vitro or human trial used to determine the mentioned action.

→ I added these in the section on Basic studies of HST.

Also when discussing western medicines you should state which medicines.

→ I added the medicines in conclusion.